Sales forecasting for retail stores using hybrid neural networks and sales-affecting variables

Mansur Saad 1
Sattar Kashif 2 kashif@uaar.edu.pk
Hosseini Seyed Ebrahim 1 seyedh@whitecliffe.ac.nz
http://orcid.org/0009-0003-4574-8944 Pervez Shahbaz 1
http://orcid.org/0000-0003-3719-2387 Ahmad Iftikhar 3
http://orcid.org/0000-0001-8062-3301 Saleem Kashif 4
Zohier Elhendi Ahmed 5
1 Whitecliffe , Auckland , New Zealand
2 University of Arid Agriculture Rawalpindi , Rawalpindi, Panjab , Pakistan
3 Faculty of Computing and Information Technology, King Abdul Aziz University , Jeddah , Saudi Arabia
4 Center of Excellence in Information Assurance, DSR, King Saud University , Riyadh , Saudi Arabia
5 Science Technology and Innovation Department, King Saud University , Riyadh , Saudi Arabia
Lara Juan
Electronic publication date: 2025 Sep 11
Publication date: 2025
Volume: 11
Electronic Location ID: e3058
Received 2025 Mar 21; Accepted 2025 Jun 30
Copyright: © 2025 Mansur et al.
Copyright year: 2025
Copyright holder: Mansur et al.
License: This is an open access article distributed under the terms of the Creative Commons Attribution License, which permits unrestricted use, distribution, reproduction and adaptation in any medium and for any purpose provided that it is properly attributed. For attribution, the original author(s), title, publication source (PeerJ Computer Science) and either DOI or URL of the article must be cited.
License URL: https://creativecommons.org/licenses/by/4.0/

Keywords: Sales forecasting, Retail store, Deep learning, CNN, LSTM, Mutivariate dataset, Hybrid neural network

Funding: King Saud University, Riyadh, Saudi Arabia ORF-Ctr-2025-4 Researchers Supporting Project RSPD2025R697 This work was supported by King Saud University, Riyadh, Saudi Arabia, through the Ongoing Research Funding program (ORF-Ctr-2025-4), King Saud University, Riyadh, Saudi Arabia formerly known as the Researchers Supporting Project under Grant No. RSPD2025R697. The funders had no role in study design, data collection and analysis, decision to publish, or preparation of the manuscript.

==============================
Accurate sales forecasting is vital for balancing demand and supply and enhancing profitability in the retail sector. Deep learning (DL) models have shown promise in this area; however, most either handle temporal or spatial patterns in isolation. Moreover, many studies rely on synthetic datasets or omit critical contextual variables, reducing real-world accuracy. This study proposes a hybrid convolutional neural network (CNN)-long short-term memory (LSTM) model for retail sales forecasting using real-world data enhanced with environmental and demographic variables in term of holidays, salary days, protests, and weather conditions. CNNs capture spatial patterns, while LSTMs model temporal dependencies, making the hybrid architecture well-suited for multivariate forecasting tasks. Our model demonstrates a significant improvement in predictive performance, achieving a mean absolute percentage error (MAPE) of 4.16%, outperforming traditional and standalone neural models. By incorporating external factors, the proposed approach enables more reliable forecasting and supports informed decision-making in retail operations.

Introduction

Sales forecasting predicts future sales based on past performance, market trends, and economic conditions. In the recent past, scholars have been working hard to improve the accuracy of sales prediction by applying various artificial intelligence (AI) techniques. The findings in a study by Ahmadov & Helo (2023), reveal that AI models can provide up to 35% better accuracy than the traditional methods. In another study by Sidabutar & Firmansyah (2023), authors compare the forecasting accuracy of linear regression, Autoregressive Integrated Moving Average (ARIMA), and neural network models for sales prediction. The results demonstrate that the neural network model performs better in terms of root mean square error (RMSE), outperforming both linear regression and ARIMA models. Inaccurate forecasting can result in stockouts or overstock, directly affecting customer satisfaction and profitability. According to Huang & Rust (2022), the core of AI lies in three fundamental concepts: neural networks, deep learning (DL), and machine learning. These concepts are driving advancements in software development, data analysis, and natural language processing. Neural network models tend to be more advantageous than traditional models in terms of predictive accuracy (Ghatora et al., 2024; Rawat, Hosseini & Pervez, 2023). They are capable of identifying dynamic non-linear trends and seasonal patterns, as well as the relationships between these trends and patterns. These facts convinced us to choose the artificial neural network (ANN) technique for sales prediction and we selected data from the retail store to apply ANN.

To acquire accurate sales forecasting for retail businesses, there is much room for improvement in finding an optimal AI solution. This research is motivated by the challenge of finding an optimal ANN-based solution for sales forecasting. This research is motivated by the pressing challenge of finding an optimal ANN technique for sales forecasting in the retail industry while addressing the obstacles and complexities that hinder its seamless adoption. Numerous studies have highlighted some AI models (DL and ANN) in retail; however, there is a notable gap in comprehensively examining how these techniques contribute to sales enhancement in retail marketing.

This study aims to improve retail sales forecasting through a hybrid deep learning model that combines convolutional neural networks (CNN) and long short-term memory (LSTM) networks. In addition to sales data, we incorporate external demographic and environmental variables such as holidays, salary days, protests, and weather conditions that have been shown to influence consumer behavior significantly (Haque, 2023; Biswas, Sanyal & Mukherjee, 2023; Ahmadov & Helo, 2023).

Several studies have explored AI techniques for sales forecasting. For instance, Ahmadov & Helo (2023) showed that deep neural networks outperform traditional models in capturing non-linear sales patterns, achieving up to 35% greater accuracy. Similarly, Sidabutar & Firmansyah (2023) found that neural networks produced lower RMSE than both ARIMA and linear regression. However, many existing models rely on synthetic or limited datasets and fail to incorporate real-world contextual variables, limiting their applicability in practical retail settings (Silva et al., 2021; Kumar et al., 2023). Some models perform well on time-series data, while others are better suited for spatial or contextual features but few address both effectively.

This gap presents an opportunity to develop a model that is not only accurate but context-aware. To this end, we use a publicly available real-world dataset from Kaggle (ITEXPERTFSD, 2024), enriched with weather and event-based data linked to the store’s geographic location. Unlike prior work such as Biswas, Sanyal & Mukherjee (2023), which focused on online customer reviews, our research emphasizes physical retail behavior, which is more directly affected by external factors like weather, holidays, and regional disruptions.

We also perform a comprehensive review and comparative analysis of recent AI-based sales forecasting approaches, identifying key methods, limitations, and areas for improvement. The proposed hybrid model is validated using actual historical data, demonstrating improved performance and practical utility for forecasting in retail environments, the key contributions of this study are: - Conducts a structured literature review and presents a comparative analysis of recent AI-based sales forecasting approaches.

- Proposes a novel hybrid CNN–LSTM model tailored for multivariate retail sales forecasting.

- Integrates environmental (rainfall, temperature) and demographic (holidays, salary days, protests) variables into the forecasting pipeline.

- Demonstrates superior forecasting accuracy over standalone models (ANN, CNN, LSTM), achieving a mean absolute percentage error (MAPE) of 4.16% on real retail data.

- Utilizes publicly available datasets (Kaggle + weather) to ensure transparency and reproducibility.

- Provides a scalable and adaptable framework for other retail stores and geographic regions.

The remainder of this article is organized as follows. ‘Related Work’ presents a review of related work on AI-based sales forecasting. ‘Methodology’ describes the dataset, preprocessing, and feature engineering. ‘Results’ explains the architecture and implementation of the proposed hybrid CNN–LSTM model. ‘Discussion’ provides performance evaluation and comparative analysis. Finally, ‘Conclusions’ concludes the study and outlines directions for future research.

Related work

This section reviews recent literature on AI-based approaches to sales forecasting, including models based on machine learning (ML), ANN, CNN, LSTM, and ensemble techniques. Several studies also propose hybrid models that combine multiple AI methods—or integrate AI with domain knowledge—to enhance forecasting accuracy. The following sub-sections detail these developments across different model categories.

Sales forecasting using machine learning

Recent machine learning (ML) methods help vendors forecast sales using event-based and contextual data, aiding inventory and profit management (Beese & Fahse, 2023). Although no model guarantees perfect accuracy, combining approaches and continuous tuning improves performance. Classical methods like ARIMA and Seasonal Autoregressive Integrated Moving Average (SARIMA) rely on statistical smoothing and are widely used for market-level forecasts (Eglite & Birzniece, 2022). However, global models such as RNNs, enhanced with covariates (e.g., time, price, events), offer better accuracy in handling interrelated time series, with performance gains of 1.76% MRMSSE and 6.47% MMASE (Ramos & Oliveira, 2023). Comparative evaluations show ML methods, including ANNs, outperform traditional approaches, especially when integrating both endogenous and exogenous variables (Martins & Galegale, 2023).

Further analysis of models like Random Forest, Linear Regression, XGBoost, and Keras DNNs shows prediction accuracy improves with product attributes and customer information (Rajasree & Ramyadevi, 2024). In fashion retail, deep neural networks (DNNs) combined with data mining were used to forecast new item sales based on historical patterns and expert input (Loureiro, Miguéis & Da Silva, 2018). Using separate seasonal datasets and grid search tuning, models achieved over 67% R2. This suggests future research should integrate richer variables and external factors for improved sales forecasting outcomes.

Sales forecasting using artificial neural network

Recent ANN variants have enhanced retail sales forecasting accuracy. One study showed that integrating macroeconomic indicators like Consumer Price Index (CPI), Inventory Control System (ICS), and unemployment rates into ANN models significantly improved prediction performance (Haque, 2023). This highlights the importance of including broader economic variables to better reflect market dynamics. To address the limitations of traditional forecasting methods such as long execution time and low accuracy Zhao, Tian & Wang (2023) introduced the QLBiGRU model, combining Q-Learning and bidirectional gated recurrent unit (BiGRU), achieving better performance through automatic parameter optimization.

Intermittent sales, marked by irregular order arrivals, pose challenges for prediction. To address this, Ahmadov & Helo (2023) proposed a DNN with multi-layered architecture capable of capturing complex patterns, improving forecasting accuracy by up to 35% compared to classical models like ARIMA and Croston’s method. Using data from 17 sellers and 3,000 orders, the study validated DNN’s applicability across diverse retail environments. In supply chain forecasting, a hybrid SVM and ANN approach was applied for the food retail sector to predict future demand (Hussain & Haroon, 2018).

Another model by Silva et al. (2021) employed Ensemble Empirical Mode Decomposition (EEMD) with Bayesian Regularization ANN, Cubist Regression, and SVR. The model decomposed historical sales into intrinsic components and trained them individually using BRNN, Cubist, and SVR with a Radial Basis Function kernel. The ensemble results EEMD–BRNN, EEMD–CUBIST, and EEMD–SVR proved highly effective for daily sales forecasting with MAPE as low as 18.85.

Customer sentiment also impacts sales outcomes. A study by Biswas, Sanyal & Mukherjee (2023) analyzed reviews from Amazon (https://www.amazon.in/) and Snapdeal (https://www.snapdeal.com/) using ANN and IBM SPSS (IBM Corp., Armonk, NY, USA) (Al-Imam, 2019). Sentiments classified by a 1–5 scale influenced the sales forecast model, showing the impact of customer feedback. The authors suggested extending the model to include demographic variables and integrate both online and offline feedback for improved predictions.

Caglayan, Satoglu & Kapukaya (2020) used ANN for sales forecasting in a Turkish retail chain by analyzing historical sales, pricing, promotions, customer footfall, and weather. The model identified optimal network architecture and achieved reliable forecasting. Regression analysis was also applied for comparison. This multi-source data integration showed the importance of contextual factors in accurate forecasting.

Sales forecasting using LSTM and ensemble learning

Ensemble methods are vital for improving sales forecasting accuracy by combining multiple models to offset individual biases. Techniques like voting regressors have shown notable performance gains, particularly with large datasets (Alice, Andrabi & Jha, 2023). A study using XGBoost, LGBM, and CatBoost demonstrated their effectiveness in retail forecasting, emphasizing their unique strengths (Alice, Andrabi & Jha, 2023). Accurate forecasts are essential for optimizing supply chains and enhancing customer satisfaction (Kilimci et al., 2019). Traditional models often struggle with complex sales data, prompting exploration into deep learning (DL) techniques.

To address data limitations, deep transfer learning has been applied, where knowledge from one task improves forecasting on another with limited data (Erol & Inkaya, 2023). Two ensemble DL methods bagged and stacked were introduced. Bagging combines predictions from long short-term memory-transfer learning (LSTM-TL) models trained on different data subsets to reduce variance. Stacking merges multiple LSTM-TL outputs through a higher-level model to capture complex relationships (Erol & Inkaya, 2023). The stacked method showed superior accuracy compared to bagging due to its deeper learning capacity.

These techniques help mitigate overfitting and improve model generalization by leveraging diverse architectures. Another integrative ensemble model combined weighted XGBoost, ARIMA, and Holt-Winters to reduce overfitting and information leakage (Aun et al., 2022).

Additional studies also favor ensemble learning. Sajawal et al. (2022) showed that XGBoost outperformed traditional regressors, with MAE = 0.516 and RMSE = 0.63. Similarly, Kumar et al. (2023) found that tuning XGBoost with RandomizedSearchCV yielded better results than multiple classic ML models.

Sales forecasting using CNN and hybrid neural networks

A hybrid model was proposed to improve demand forecasting per SKU by selecting the best two among five regression methods (LR, PR, RF, SVR, GBR), achieving a MAPE of 7.74% (Taparia et al., 2024). However, it focused mainly on price-related features and lacked broader sales influencers like holidays or weather. Deep learning integration using CNN and LSTM has also been explored through stacked and parallel hybrid models, achieving MAPEs of 14.15% and 14.76% respectively on retail data (Castro Moraes, Yuan & Chew, 2024), though limited by univariate inputs.

Other hybrid strategies combine trend and residual learners (e.g., LR, ElasticNet, RF, XGB) to boost accuracy, such as the Lucy model which supports model saving for efficient reuse (Tran, Huh & Kim, 2023). Another approach embedded exponential smoothing into a GRU-CNN-LSTM framework (ATES) to handle seasonal data, improving accuracy over baselines like WaveNet and TFT (Efat et al., 2024). This model leveraged both sequence modeling and clustering, trained on multi-year sales data.

Hybrid ARIMA-BPNN models have also incorporated content popularity (from Google search data) into sales forecasts for magazine retail, outperforming traditional models through combined trend and nonlinear pattern learning (Omar, Hoang & Liu, 2016).

Recent work in retail supply chains recommends multivariate models that include store location, weather, and promotions. A GRU model optimized via Grid Search, applied to Rossmann store data, showed stronger forecast accuracy, although it lacked store closure information (Qureshi et al., 2024).

Collectively, studies show that while many AI models (ANN, ML, LSTM, CNN, RNN, hybrids) exist for retail forecasting, most use artificial or incomplete real datasets that ignore crucial variables like environmental or demographic data. Since no single model handles both spatial and temporal dependencies well, there is a clear need for hybrid models that integrate these factors without compromising user privacy. Recent comparisons confirm CNN-BiLSTM hybrids outperform standalone models like ARIMA, CNN, or LSTM in terms of RMSE and MAE (Joseph et al., 2022). Similarly, Ahmed et al. (2024) found CNN-LSTM hybrids more robust and accurate than individual DL architectures, offering a balance of accuracy and interpretability without requiring large datasets or excessive tuning.

By analysing the solutions provided in the available literature above, we have come up with a novel and optimal solution based on the hybrid (CNN and LSTM) model and by considering some environmental and demographic variables, to improve forecasting sales. In Table 1, some research studies are listed that we found closest to the sales forecasting domain and are using AI models to improve sales forecasting accuracy. A comparative analysis of these research studies is done on the basis of MAPE value, as we will be using MAPE as our base evaluation metric for comparing the results of our proposed solution with these existing research studies.

Table 1 Comparison of related work with the proposed model.

Article reference	Forecasting model	Dataset type	External variables	MAPE	
Biswas, Sanyal & Mukherjee (2023)	Feed-forward ANN (customer feedback)	E-com reviews (Amazon, Snapdeal)	None	–	
Haque (2023)	LSTM + macro vars	Sales + macro indicators	CPI, ICS, Unemployment	–	
Ahmadov & Helo (2023)	DNN + Poisson Exp. Dist.	eBay sellers (17)	None	7%	
Silva et al. (2021)	EEMD (BRNN, Cubist, SVR)	Rossmann (7 months)	None	18.85%	
Kumar et al. (2023)	XGBoost + tuning	Kaggle retail dataset	None	–	
Taparia et al. (2024)	Hybrid (LR-PR, RF-LR)	1,000 SKUs, 2 yrs	Price, Competitor price	7.74%	
Castro Moraes, Yuan & Chew (2024)	S-CNN-LSTM (stacked/parallel)	125 M retail records (5 yrs)	None	14.15%	
Proposed	Hybrid (CNN+LSTM)	Kaggle + Weather + Events	Rain, Temp., Holidays, Salary, Events, Midweek, Protest	4.16%	

Methodology

Our proposed methodology for forecasting sales is mainly based on a combination of CNN and LSTM model, which makes use of sales data collected from the retail store for the customers visited physically (public data available at Kaggle) (ITEXPERTFSD, 2024), along with demographic and environmental factors. The methodology and its components are explained in the following sections.

CNN model

CNNs, while originally developed for image processing, have also shown promise in identifying patterns within structured data. Their strength lies in modeling complex, non-linear relationships between static input features such as promotions, store locations, or environmental indicators and sales outcomes. Although CNNs can detect local trends in time series data, they are limited in capturing long-term temporal dependencies due to the absence of a recurrent structure. In our proposed model, CNN is used to extract short-term feature patterns that contribute to sales variation.

LSTM model

LSTM networks are well-suited for modeling sequential data and capturing temporal dependencies, such as seasonality or recurring demand shifts in sales. They retain information across time steps, allowing the model to learn from past patterns that influence future outcomes. However, LSTMs can be computationally intensive and sensitive to noise if data is sparse or unrefined. In our hybrid model, LSTM complements CNN by addressing the temporal aspects of sales forecasting, particularly where historical trends and event-driven fluctuations are significant.

Environmental and demographic factors

Retail sales forecasting is influenced by both internal and external variables. Beyond the core fields (date, sales, and open/closed status) in our Kaggle dataset (ITEXPERTFSD, 2024), we incorporated additional environmental and demographic factors:

(A) Environmental Variables

Based on sales drops observed on certain days, we matched weather data from the city (Crossing, 2024) and defined: EffectiveRain: Days with precipitation ≥20 mm were labeled as rainy. These often corresponded with decreased sales.

EffectiveTemperature: Days with temperatures ≥40 °C were marked as hot days, also associated with reduced customer activity.

(B) Demographic Variables

Sales fluctuations also aligned with specific calendar or social events. We categorized them as: PositiveEvents: Includes holidays, salary days, store promotions, and national/religious occasions typically boosting sales.

NegativeEvents: Includes mid-week days (notably Wednesday), protest days, and partial closures linked with lower sales activity.

Proposed model and data overview

To effectively capture both temporal patterns and short-term fluctuations in retail sales, we employed a hybrid CNN-LSTM architecture (Fig. 1). LSTM components handle sequential dependencies such as trends and seasonality, while CNN layers help detect local patterns like sales spikes due to promotions or events. This complementary structure improves forecast precision, especially in volatile sales environments.

Figure 1 High level diagram for proposed model.

The model pipeline begins with data acquisition, followed by preprocessing and feature extraction. After transformation, inputs pass through CNN and LSTM layers sequentially, with the final dense layer producing the sales forecast. The model’s performance is evaluated using MAE, RMSE, and MAPE metrics.

Data collection and analysis

We used a publicly available Kaggle dataset containing 6 months of daily sales data from a retail store in Faisalabad, Pakistan (ITEXPERTFSD, 2024). To enrich the dataset, we integrated external demographic and environmental variables sourced from local public data. Daily Sales Data: Includes total daily sales figures for the store.

Demographic Factors: Includes events such as holidays, promotions, salary days, protests, and partial closures that influence consumer activity.

Environmental Factors: Incorporates temperature and precipitation data to capture weather-related impacts on store traffic and sales.

These variables enable the hybrid model to learn from both internal and external sales influencers, improving its ability to forecast demand under varying real-world conditions.

Results

We have applied our sales forecasting hybrid model which is a combination of LSTM and CNN on the available sales dataset using Python programming language. Once the ANN model is applied to the dataset, we analyse the results by different metrics like mean absolute error, RMSE and MAPE. The implementation of our proposed hybrid model aims to forecast monthly sales for the retail store by making use of the last 6 months’ sales data.

Tools and technologies and computing infrastructure

The proposed CNN-LSTM hybrid model was developed and tested using Python and standard machine learning libraries on Google Colab, which offers cloud-based GPU acceleration. The tools used supported data preprocessing, model training, evaluation, and visualization. Key frameworks included TensorFlow/Keras for deep learning and Scikit-learn for preprocessing and performance metrics. The full software stack and system specifications are summarized in Table 2.

Table 2 Software libraries and computing environment.

Component	Details	
Platform	Google Colab (cloud-based Jupyter environment with GPU support)	
OS & Python	Ubuntu 18.04 backend, Python 3.10.12	
CPU & RAM	Intel Xeon @ 2.30 GHz, 12.72 GB RAM	
GPU	NVIDIA Tesla K80 or T4, 12–16 GB memory	
TensorFlow/Keras	TensorFlow 2.13.0, Keras 2.13.1—For building and training DL models	
Pandas/NumPy	Pandas 1.5.3, NumPy 1.23.5—Data handling and transformations	
Matplotlib/Seaborn	Matplotlib 3.7.1, Seaborn 0.12.2—Visualization tools	
Scikit-learn	1.2.2—Preprocessing, splitting, and performance evaluation	

Baselining models for comparison

To assess the performance of the proposed model, we implemented two baseline models (LSTM & CNN) for comparison. Each model was trained and evaluated on the same dataset using standardized preprocessing techniques. Performance was measured using common regression metrics including MAE, RMSE, and MAPE.

By comparing the proposed hybrid model’s results against these baseline models, the study demonstrates a significant improvement in predictive accuracy, especially when external influencing variables are included. This comparative analysis provides stronger support for the effectiveness of the hybrid neural network model in retail sales forecasting.

Data pre-processing

Data preprocessing is a crucial step before feeding the data into the models. Our dataset consists not just of daily sales data but also some other demographic and environmental data related to the retail store location. So, the following preprocessing steps were carried out to make data suitable for our model to forecast sales accurately: Handling Qualitative Data: Qualitative data like holidays, national events/festivals and stores open/closure/partially closed, hot weather, effective rain and protests are required to be converted into numerical format. It helps train the model to get the best forecasting results.

Normalisation/Scaling: The data was normalised using Min-Max Scaling to ensure that all input features are within a similar range, which improves model performance.

Feature Engineering: Additional features such as (lagged sales values, moving averages, holiday flags) were created to capture trends and seasonality. The time series was also decomposed into (trend, seasonal, and residual components) for better model interpretation.

Data Availability: The final dataset consisting of sales data obtained from Kaggle, publicly available weather data and public holidays data is consolidated together as one Dataset (CSV file), which is available at Mansur (2025).

Proposed model implementation details

After analyzing the sales data, we implemented a hybrid CNN-LSTM model to capture both local (short-term) and global (long-term) patterns. The model begins with 1D convolutional layers that apply filters across the time dimension to detect temporal features such as sales spikes, dips, and short-term trends often influenced by promotions, holidays, or weather events. These filters help extract spatial patterns from time-series data by sliding over sequences of input shaped as (timesteps, features). The convolutional output is then passed to LSTM layers, which capture sequential dependencies and long-term trends. As detailed in Table 3, the dataset includes multiple external variables such as weather thresholds and event indicators which enhance the model’s ability to recognize dynamic influences on sales behavior.

Table 3 Description of sales-affecting variables.

Variable	Description	
Sale	Total sales amount in a day	
Status	Store is open or closed	
Day of week	Derived from the date field	
Effective rain	Binary indicator: 1 if precipitation >20 mm, based on observed drop in sales during heavy rain	
Effective temperature	Binary indicator: 1 if temperature >40 °C, set based on observed decline in sales during hot weather	
Positive events	Events likely to increase sales (e.g., promotions, holidays)	
Negative events	Non-weather events that reduce sales (e.g., protests, closures)	

The hybrid CNN-LSTM model was fine-tuned through empirical testing to balance performance and complexity. For the CNN component, we used 64 filters with a kernel size of 4 and ReLU activation to extract meaningful short-term patterns from the 6-month sales dataset. A MaxPooling1D layer with a pool size of 2 was applied to reduce dimensionality while retaining dominant features.

The output from the CNN layers was passed to a single LSTM layer with 50 units to capture temporal dependencies such as trends and seasonality. A fully connected dense layer followed, mapping extracted features to the final sales prediction. The output layer used a single neuron with linear activation to produce the forecasted value.

Training was conducted using the Adam optimizer and MSE as the loss function. We trained the model for 150 epochs with a batch size of 32, and applied a 10% validation split to monitor performance. Dropout and additional stacked LSTM layers were omitted after experimentation showed that a single-layer configuration yielded stable and accurate results.

All selected hyperparameters are summarized in Table 4, based on manual tuning and performance evaluation using metrics such as MAPE and RMSE.

Table 4 Final selected hyperparameters for the CNN-LSTM hybrid model.

Component	Hyperparameter	Selected value	
CNN layer	Number of filters	64	
	Kernel size	4	
	Activation function	ReLU	
	Pooling type	MaxPooling1D (pool size = 2)	
LSTM layer	Number of units	50	
	Number of layers	1	
	Return sequences	False	
Output layer	Dense units	Equal to output shape (dependent on prediction)	
Training	Optimizer	Adam (default learning rate)	
	Loss function	Mean Squared Error (MSE)	
	Batch size	32	
	Epochs	150	
	Validation split	0.1	

Model training

The model is trained using the training dataset, where historical sales data is split into sequences. Our model is trained for 150 epochs with early stopping based on validation loss to avoid over-fitting. We have tried the model with different epoch values to find an optimal number where the model performs best and overcomes under-fitting and over-fitting successfully. Results for different epoch values we tried are summarised in Table 5, where we can see that the model learns from these sequences to predict better future sales with MAPE 4.16%, for epochs value of 150. The input sequence length was set to 32 days to capture recent trends.

Table 5 Comparison of Epoch values.

Epoch value	MAE	RMSE	MAPE	
80	31,150	57,032	11.57%	
100	21,326	45,661	7.89%	
150	11,165	26,200	4.16%	
200	17,281	42,817	6.05%	

Analysis of solution evaluation and testing

We have evaluated our proposed hybrid model using various performance metrics to assess forecasting accuracy. The dataset was split into training and test sets using an 80–20 ratio. The following metrics were used for evaluation and analysis: MAE: Measures the average magnitude of the errors in a set of predictions.

(1) MAE=1n∑i=1n|yi−y^i|

RMSE: Gives more weight to large errors, highlighting model performance on outliers.

(2) RMSE=1n∑i=1n(yi−y^i)2

MAPE: Useful for evaluating the percentage accuracy of the forecast compared to actual values.

(3) MAPE=1n∑i=1n|yi−y^iyi|×100

Challenges and limitations

Since the start of the implementation phase for finding the optimal sales forecasting model, we have encountered multiple challenges which were resolved as we proceeded with more investigation and analysis of the dataset.

Spikes in sales data

The LSTM model is suitable for time series data forecasting and finds out the points where the behaviour is entirely different. Therefore, after analysing the sales data and passing it through the LSTM sales forecasting model, we came to know that our sales dataset has spikes (increases or decreases in sales amount) on some specific days of the month. So, after conducting further investigation and considering the demographic and environmental factors of the store location, we found out that sales numbers spiked due to the presence of these factors, like having more sales on a holiday or promotion day and fewer sales on a rainy day. Figure 2, shows a snapshot of the spikes in the original sales data. On the x-axis we have dates in increasing order and on the y-axis we have sales in PKR.

Figure 2 Sales fluctuations showing all specified spikes and drops including the dramatic drop to zero on 10-Apr, along with spikes on 11-Mar (700,000), 9-Apr (950,000), 17-Jun (850,000), and 1-Sep (500,000).

Limitations of single-model sales forecasting

We began by applying individual deep learning models to the sales data to evaluate their forecasting performance. First, the ANN model was used. As a feed-forward neural network, ANN processes inputs in one direction and lacks temporal awareness. Even after integrating weather and demographic features, its MAPE remained high at 19.74%, indicating poor accuracy.

Next, the CNN model an advanced variant of ANN with convolutional layers was applied. While performance improved over ANN due to local feature extraction, its MAPE of 13.90% (with all features) was still suboptimal.

We then tested the LSTM model, which is specifically designed for sequence modeling and temporal dependencies. LSTM yielded better results with a MAPE of 10.39%, showcasing its ability to capture long-term patterns in sales data.

Table 6 summarizes the comparative results of the three models across three dataset variations: sales-only, sales with weather, and sales with weather plus demographic features.

Table 6 Performance comparison of ANN, CNN, and LSTM models on different input configurations.

Model	Input data	MAE	RMSE	MAPE	
ANN	Sales only	62,817.61	79,389.71	25.63%	
	+ Weather	58,112.11	71,411.91	22.34%	
	+ Weather + Demographic	53,464.81	66,565.89	19.74%	
CNN	Sales only	47,752.50	62,114.03	18.95%	
	+ Weather	36,417.95	47,935.79	14.13%	
	+ Weather + Demographic	37,004.71	49,582.81	13.90%	
LSTM	Sales only	43,299.81	65,350.01	16.21%	
	+ Weather	29,465.56	38,860.54	11.35%	
	+ Weather + Demographic	27,403.56	36,945.79	10.39%	

From these results, we observe that while each successive model improves forecasting accuracy, none achieves satisfactory results alone. ANN lacks temporal modeling, CNN captures spatial trends but not long-range dependencies, and LSTM handles time-series well but not spatial patterns. This motivated the development of a proposed hybrid CNN+LSTM model to leverage both spatial and temporal features for improved sales forecasting performance.

Rationale for the hybrid model

Initial experiments showed that while CNN outperformed ANN due to its feature extraction capabilities, LSTM was superior for modeling temporal dependencies. Neither model alone yielded optimal results, especially in handling the complex and dynamic patterns in sales data. This led us to adopt a hybrid CNN-LSTM approach, capable of capturing both spatial and temporal aspects of sales influenced by weather and demographic factors.

Novelty of dataset and limitations in comparison

Most existing works use synthetic or univariate datasets, often excluding environmental and demographic variables. In contrast, our study uses a real-world dataset with location-based attributes and carefully selected external factors, which enhances forecasting accuracy. Due to the lack of similar datasets in prior research, direct comparisons are limited. Nonetheless, our model demonstrated lower MAPE values compared to benchmarks from the literature, showing the effectiveness of our approach.

Ethical considerations

The dataset used in this study is publicly available from Kaggle (ITEXPERTFSD, 2024), and contains no personally identifiable information (PII). To ensure privacy, the store identity is anonymized, and no efforts were made to de-anonymize or link data to external sources. As no human subjects were involved, formal IRB approval was not required. All data handling adhered to principles of accuracy, reliability, and integrity.

Discussion

The application of the hybrid CNN-LSTM model for sales forecasting proved to be highly effective in predicting future sales based on the time-series (temporal) and spatial data. By leveraging both the CNN layers to capture short-term patterns and LSTM layers to model long-term dependencies, the model is able to provide accurate and reliable sales forecasts.

Model findings and performance evaluation

The performance of the hybrid CNN-LSTM model was evaluated using standard metrics such as MAE, RMSE, and MAPE as shown in Table 7, which are commonly used to assess the accuracy of time-series forecasting models.

Table 7 Sales forecasting using demographic and environmental factors.

Dataset	MAE	RMSE	MAPE	
Sales data with environment and demographic factors	11,165	26,200	4.16%	

MAE: The MAE value of 11,165 indicates the average magnitude of the forecasting error in terms of actual sales units. This suggests that, on average, the predicted sales values deviate from the actual sales by approximately 11,165 units.

RMSE: The RMSE value of 26,200 reflects the overall error magnitude with more weight given to larger deviations. Although slightly higher than MAE, this value emphasises that larger forecasting errors occur occasionally but are relatively infrequent.

MAPE: The MAPE value of 4.16% indicates that the model’s predictions, on average, are off by only 4.16% of the actual sales values. This percentage is quite low, demonstrating the model’s high accuracy and efficiency in forecasting sales over time.

Critical analysis and comparison

In this section, we critically analysed and compared the performance of the proposed hybrid CNN-LSTM model. Also discussed are the benefits of the hybrid model and the impact of environmental and demographic factors.

The benefit of using hybrid sales forecasting model

The relatively low MAPE of 4.16% as shown in Table 7 suggests that the proposed hybrid model with multivariate data is capable of making highly accurate forecasts. It is very useful even in a business environment where small fluctuations in sales can occur due to external factors like promotions, weather conditions, or holidays/events. The hybrid model successfully captures both short-term variations and long-term trends. The CNN layers efficiently detected local patterns, such as sales spikes due to promotional campaigns or holidays, while the LSTM layers captured temporal dependencies, allowing the model to learn from past trends and project them into the future.

The performance of the model as indicated by the MAE and RMSE metrics confirms that the model is capable of making reliable sales forecasts, even with some occasional larger deviations that are reflected in the RMSE value.

Impact of demographic factors

Incorporating demographic factors like store status, holidays, events, promotions and protests, significantly improved the hybrid CNN-LSTM model’s performance in forecasting sales. These variables provide valuable context by introducing influential external factors that affect consumer behaviour, thereby increasing the model’s ability to accurately predict sales patterns.

Sales patterns during holiday seasons or special events often show significant deviations from typical trends. By including holiday data, the model could capture these spikes, which are essential for accurately forecasting peak sales periods. Similarly, promotions can create temporary sales increases that do not follow regular patterns. By incorporating promotion data, the model was better equipped to account for short-term sales surges, particularly for time-limited campaigns, making its forecasts more adaptable to marketing strategies. We have identified these events as positive events as they are increasing the sales number for the store. There are some days of the week where a decrease in sales data can be seen and that day is identified as the middle day of the week in our dataset. We named that middle day (Wednesday) and protests as a negative event in our dataset and trained our model accordingly on these features.

Influence of environmental factors

Weather conditions including rain and hot days have been shown to influence consumer purchasing behaviour. For instance, in hot weather, people avoid coming out for shopping and this behaviour decreases retail sales. Similarly, effective rainy days affect store foot traffic. Adding these environmental factors allowed the model to anticipate demand shifts related to weather changes, providing a more nuanced and precise sales forecast.

Influence of environmental and demographic factors

We can find the collective impact of adding demographic and environmental factors by looking at the values of MAPE in Table 8.

Table 8 Sales forecasting using demographic and environmental factors.

Dataset	MAE	RMSE	MAPE	
1st run: with only sales data	21,667	39,216.71	8.02%	
2nd run: sales data with environment factors	21,326	45,661	7.89%	
3rd run: sales data with demographic factor	11,979	24,167	5.14%	
4th run: sales data with environment and demographic factors	11,165	26,200	4.16%	

When we run the proposed model with only univariate sales data, the MAPE value is 8.02%. In the second run, we modified our dataset and code by adding environmental factors like precipitation and temperature, the MAPE value is 7.89%. This reduction in error shows improvement in results, which proves that these variables are affecting sales forecasts and should be considered while forecasting future sales. In the third run, we removed environmental factors and added some demographic factors like store open/close/partially closed, promotions in sales, holidays, events and protests. We can see from the results that these factors or variables are more effective on sales and by considering these variables we can reduce the MAPE error to 5.14%. Then, finally, we ran our proposed model by considering all variables together. These factors contributed to the hybrid CNN-LSTM model achieving a low MAPE of 4.16%, indicating that the model’s predictive accuracy was notably enhanced by the inclusion of both demographic and environmental features. This integration of external variables made the model more versatile and responsive to the various dynamic influences on consumer behaviour, highlighting its effectiveness as a sales forecasting tool in complex, real-world conditions. The performance comparison between the first run (using only sales data) and the third run (incorporating demographic factors) as shown in Fig. 3, reveals a significant improvement in forecasting accuracy. As shown in the graph, the MAE dropped from 21,667 to 11,979 and the RMSE decreased from 39,216.71 to 24,167, indicating a substantial reduction in both absolute and squared error. Moreover, the MAPE improved from 8.02% to 5.14%, demonstrating better relative prediction accuracy. These improvements suggest that integrating demographic variables into the input dataset enhanced the model’s ability to capture patterns influencing sales behavior, leading to more reliable and accurate forecasts.

Figure 3 Comparison of with and without demographic factors.

The most representative 30 entries from the 60-day forecast output of the model’s 4th run, along with corresponding actual sales, are shown in Table 9.

Table 9 Sales forecast 4th run testing results.

Sr.#	Date	Forecast	Actual	Sr.#	Date	Forecast	Actual	
1	20/07/2024	265,987	267,990	31	19/08/2024	204,762	202,740	
5	24/07/2024	177,357	175,229	35	23/08/2024	216,116	220,167	
10	29/07/2024	161,950	159,432	40	28/08/2024	183,005	177,850	
12	31/07/2024	238,606	236,590	42	30/08/2024	212,852	210,000	
14	02/08/2024	411,558	411,546	44	01/09/2024	492,786	492,061	
16	04/08/2024	358,181	354,847	46	03/09/2024	271,233	269,059	
18	06/08/2024	306,314	301,310	48	05/09/2024	345,141	371,018	
20	08/08/2024	233,098	228,803	50	07/09/2024	245,036	321,005	
22	10/08/2024	275,955	278,283	52	09/09/2024	219,303	221,824	
24	12/08/2024	238,543	235,975	54	11/09/2024	228,333	203,174	
26	14/08/2024	284,181	281,416	56	13/09/2024	197,255	165,445	
28	16/08/2024	233,950	234,655	58	15/09/2024	198,471	308,632	
30	18/08/2024	293,234	294,257	60	17/09/2024	278,710	284,303	

Actual and forecasted values are compared and plotted graphically as shown in Fig. 4. On x-axis dates are in increasing order, whereas on y-axis sales are in PKR. We can easily see the difference between both actual and forecast values.

Figure 4 Actual vs. forecast output comparison.

After applying our proposed sales forecasting model on the dataset consisting of 6 months of sales data with all demographic and environmental factors, we can infer that model performed well by looking at the forecasted values that are matching with the actual sale values. The results in Fig. 5 have shown that the applied forecasting model has successfully picked spikes in original sales values by getting trained with provided environmental and demographic factors.

Figure 5 Comparison graph between forecast and original data.

In Fig. 6, training loss and validation loss values are plotted where the x-axis shows the epochs of the training process and the y-axis shows the loss value. We can see a continuous decrease in training loss indicates that the model is improving its ability to fit the training data and it is learning well during the training process. On the other hand, looking at the values for validation loss shows that the model does not fit well for any unseen data, hence memorising the training data but failing to generalise to new examples. The reason for this over-fitting is the presence of sales spikes in the dataset, which are forecasted very well by the model after we added the demographic and environmental factors specifically for our dataset. This concludes that the proposed model requires the dataset to have the appropriate factors accordingly and then be fed into the model.

Figure 6 Training vs. validation loss.

Discussion on the results of different models on the same dataset

Based on our experiments, individual models such as ANN, CNN, and LSTM showed limited forecasting accuracy, even after incorporating external variables. Among them, LSTM performed better due to its ability to capture temporal patterns, but it still lacked spatial awareness.

The hybrid CNN-LSTM model, which combines convolutional feature extraction with temporal sequence modeling, showed a clear improvement. When demographic and environmental variables were integrated into this architecture, the model achieved a significantly lower MAPE of 4.16%, outperforming all baseline approaches.

These findings confirm that combining spatial and temporal learning in a hybrid framework, along with context-aware variables, substantially enhances sales forecast precision in retail environments.

Conclusions

This study proposed a hybrid CNN–LSTM model for retail sales forecasting that effectively captures both spatial and temporal sales patterns. CNN components extract localized features such as short-term trends, while LSTM layers model sequential dependencies, enabling the network to learn from historical sales behavior. This dual architecture outperformed standalone models, achieving a MAPE of 4.16% on real-world data.

Incorporating external variables such as weather conditions, holidays, salary days, and protest events further enhanced model accuracy. These environmental and demographic factors provided essential context often ignored in previous studies, allowing the model to better capture fluctuations in customer behavior.

However, the study has limitations. The dataset is limited to a single store, which constrains the model’s generalizability across regions or chains with differing consumer patterns. Future work should explore multi-store datasets, integrate region-specific variables, and assess scalability across diverse retail formats.

Additionally, the inclusion of variables like “Nearby Store Status” could further improve forecasts, as temporary closures of nearby stores significantly impact local demand. Despite data limitations, this research provides a robust framework for context-aware, data-driven sales forecasting and lays the groundwork for future developments in AI-driven retail analytics.

Supplemental Information

Supplemental Information 1 Readme.

Supplemental Information 2 Algorithms and Source Code Implementation.

Supplemental Information 3 Sales Dataset.

Supplemental Information 4 Supplemental File.

Additional Information and Declarations

Competing Interests

The authors declare that they have no competing interests.

Author Contributions

Saad Mansur performed the experiments, analyzed the data, performed the computation work, prepared figures and/or tables, authored or reviewed drafts of the article, and approved the final draft.

Kashif Sattar conceived and designed the experiments, performed the experiments, analyzed the data, performed the computation work, authored or reviewed drafts of the article, and approved the final draft.

Seyed Ebrahim Hosseini performed the experiments, analyzed the data, performed the computation work, authored or reviewed drafts of the article, and approved the final draft.

Shahbaz Pervez conceived and designed the experiments, performed the computation work, authored or reviewed drafts of the article, and approved the final draft.

Iftikhar Ahmad conceived and designed the experiments, analyzed the data, authored or reviewed drafts of the article, and approved the final draft.

Kashif Saleem conceived and designed the experiments, authored or reviewed drafts of the article, and approved the final draft.

Ahmed Zohier Elhendi performed the computation work, prepared figures and/or tables, authored or reviewed drafts of the article, and approved the final draft.

Data Availability

The following information was supplied regarding data availability:

The data is available at Kaggle: https://www.kaggle.com/datasets/itexpertfsdpk/retail-sales-dataset-for-prediction.

The source code is available at GitHub:

- https://github.com/saadmansur/sales_prediction_hybrid_ml.

- saadmansur. (2025). saadmansur/sales_prediction_hybrid_ml: Latest Release (v1.0.0). Zenodo. https://doi.org/10.5281/zenodo.15861971.

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
