# Peer review of "Sales forecasting for retail stores using hybrid neural networks and sales-affecting variables"

_PeerJ Computer Science, doi:10.7717/peerj-cs.3058_

## Round 0.1 · original submission · Minor Revisions

Please, address all the issues raised by the reviewers. In particular, it is important to include a literature review section and explain variables used, hyperparameter tuning and why you treated variables as qualitative. Also, give more details about data pre-processing, model design, and statistical testing, as well as about potential ethical aspect of this study.

·

Basic reporting

Limited references for central claims (for example, "traditional models are inferior to neural networks" lack concrete references). Include references for comparative statements (for example, refer to studies establishing ANN superiority over regression).

Lack of definitions and theorems: Provide clear definitions for all technical terms and elaborate on key methodologies in separate subsections or footnotes as necessary.

Objectives are presented in bullet points (Lines 56–64) instead of being incorporated into the narrative. Reorder objectives in a narrative structure within the Introduction.

The manuscript does not have a literature review section, which needs to be relocated to an independent "Related Work" section. Rearrange sections: Precede Methods with a standalone Literature Review.

Experimental design

No mention of ethical guidelines (e.g., anonymizing data, adherence to Kaggle's terms). Include a subsection on Ethical Considerations (sourcing of data, privacy).

Limited technical detail in describing computational infrastructure (e.g., hardware, version of the framework). Identify software libraries (e.g., TensorFlow version, PyTorch version) and the machines trained on.

No data preprocessing description (e.g., missing values handling, normalization). Add a subsection, Data Availability, that includes preprocessing stages as well as hyperparameters. Offer a public link to the dataset or make it available in supplementary material.

No statistical tests for checking the significance of gains against baselines (for example, Diebold-Mariano). Compare with baselines (for example, SARIMA, Prophet) and explain metric selection.

No ablation study to separate the effects of demographic/environmental variables. Add a Model Ablation subsection to estimate variable contributions.

Validity of the findings

Add a section clearly outlining the new contributions of the study and how they improve existing methodologies for sales prediction.

Provide more information about the experimental setup, such as baseline models for comparison, to better support the findings.

Expand on the limitations of this study, e.g., the scope of data or possible biases. Next, outline some possible directions for future research that could extend from this work.

No explicit motivation for the replication of the CNN-LSTM hybrid architecture (e.g., why this combination rather than newer architectures such as Transformers?). Insert a Replication Rationale section justifying the choice of the hybrid model.

Conclusions do not mention limitations (e.g., single-store dataset, untested scalability). Rewrite the Conclusion with limitations (e.g., geographic narrowness) and future work (e.g., testing on multi-store datasets).

Reviewer 2 ·

Basic reporting

The manuscript is written in clear and professional English throughout, with no ambiguity in presentation. The introduction effectively establishes the context and motivation, supported by relevant and well-referenced literature. The paper’s structure aligns with journal standards and disciplinary norms, with any deviations serving to enhance clarity. All formal results are well defined, with precise terminology and comprehensive proofs where appropriate. Overall, the manuscript mostly satisfies all criteria. However, there are some minor revisions suggested about the section structure.
1)A dedicated Literature Review section should be included to enhance the clarity and structure of the manuscript. At present, the review of related works is embedded within the Research Methodology section, which obscures the conceptual foundation and contextual positioning of the study.

2) It is recommended that an additional column be incorporated into Table 1 to explicitly list the external variables examined in each referenced study. This will facilitate clearer comparisons of external variable considerations across the reviewed literature. Furthermore, a final row representing “This research” should be added to Table 1 to enable a direct and transparent comparison with prior studies.

3) Once the Literature Review section is established, the authors should incorporate additional relevant studies. Expanding the literature base will strengthen the academic rigor and demonstrate a more comprehensive understanding of existing work in the field.

Experimental design

The overall experimental design is clearly articulated; however, several aspects require further clarification and improvement:

1) The rationale for treating rainfall and temperature as qualitative variables (i.e., binary indicators such as "yes/no") rather than utilizing their actual quantitative values is not sufficiently justified. If the authors initially incorporated these variables in quantitative form but found that the resulting model performance was inferior to the proposed qualitative approach, the corresponding results and analysis should be reported to support this decision.

2) The definition and explanation of demographic variables remain vague. To enhance clarity and support their relevance, the authors are encouraged to provide visual evidence, such as graphs or plots, illustrating the impact of these variables on the model outcomes.

3) The discussion surrounding hyperparameter tuning is minimal. It is unclear whether the authors optimized key parameters for the CNN models—such as learning rate, batch size, activation function, kernel size, dropout rate, and regularization methods—or the LSTM models, including sequence length, number of units, learning rate, dropout rate, and the number of layers. Further elaboration on the hyperparameter selection and tuning process would improve the transparency and reproducibility of the work.

Validity of the findings

In the discussion section, it would be beneficial for the authors to provide insights regarding the relative impact of each external variable on sales. Specifically, identifying which variables have the strongest positive and negative influences would offer greater depth to the analysis and enhance the interpretability of the findings for readers.

---

## Round 0.2 · accepted · Accept

I have confirmed that authors have addressed properly the reviewers' comments. As reviewers suggested minor revisions in previous round, I consider that the manuscript can be accepted for publication.